# Impact of Insolation Data Source on Remote Sensing Retrievals of Evapotranspiration over the California Delta

**Martha Anderson [1],\*** , **George Diak [2]**, **Feng Gao [1]** , **Kyle Knipper [1]**, **Christopher Hain [3]**,
**Elke Eichelmann [4]**, **Kyle S. Hemes [4]** , **Dennis Baldocchi [4]** , **William Kustas [1]** and **Yun Yang [1]**

[1]   Hydrology and Remote Sensing Laboratory, USDA-ARS, Beltsville, MD 20705, USA;
      feng.gao@ars.usda.gov (F.G.); kyle.knipper@ars.usda.gov (K.K.); bill.kustas@ars.usda.gov (W.K.);
      yun.yang@ars.usda.gov (Y.Y.)
[2]   Space Sciences and Engineering Center, University of Wisconsin-Madison, Madison, WI 53706, USA;
      george.diak@ssec.wisc.edu
[3]   NASA Marshall Space Flight Center, Huntsville, AL 35805, USA; christopher.hain@nasa.gov
[4]   Department of Environmental Science, Policy and Management, University of California, Berkeley, CA 94720,
      USA; eeichelm@berkeley.edu (E.E.); khemes@berkeley.edu (K.S.H.); baldocchi@berkeley.edu (D.B.)
\*    Correspondence: martha.anderson@ars.usda.gov; Tel.: +1-301-504-6616

**Abstract:** The energy delivered to the land surface via insolation is a primary driver of evapotranspiration (ET)—the exchange of water vapor between the land and atmosphere. Spatially distributed ET products are in great demand in the water resource management community for real-time operations and sustainable water use planning. The accuracy and deliverability of these products are determined in part by the characteristics and quality of the insolation data sources used as input to the ET models. This paper investigates the practical utility of three different insolation datasets within the context of a satellite-based remote sensing framework for mapping ET at high spatiotemporal resolution, in an application over the Sacramento–San Joaquin Delta region in California. The datasets tested included one reanalysis product: The Climate System Forecast Reanalysis (CFSR) at 0.25° spatial resolution, and two remote sensing insolation products generated with geostationary satellite imagery: a product for the continental United States at 0.2°, developed by the University of Wisconsin Space Sciences and Engineering Center (SSEC) and a coarser resolution (1°) global Clouds and the Earth's Radiant Energy System (CERES) product. The three insolation data sources were compared to pyranometer data collected at flux towers within the Delta region to establish relative accuracy. The satellite products significantly outperformed CFSR, with root-mean square errors (RMSE) of 2.7, 1.5, and 1.4 MJ·m$^{-2}$·d$^{-1}$ for CFSR, CERES, and SSEC, respectively, at daily timesteps. The satellite-based products provided more accurate estimates of cloud occurrence and radiation transmission, while the reanalysis tended to underestimate solar radiation under cloudy-sky conditions. However, this difference in insolation performance did not translate into comparable improvement in the ET retrieval accuracy, where the RMSE in daily ET was 0.98 and 0.94 mm d$^{-1}$ using the CFSR and SSEC insolation data sources, respectively, for all the flux sites combined. The lack of a notable impact on the aggregate ET performance may be due in part to the predominantly clear-sky conditions prevalent in central California, under which the reanalysis and satellite-based insolation data sources have comparable accuracy. While satellite-based insolation data could improve ET retrieval in more humid regions with greater cloud-cover frequency, over the California Delta and climatologically similar regions in the western U.S., the CFSR data may suffice for real-time ET modeling efforts.

**Keywords:** evapotranspiration; insolation; surface energy balance; data fusion; water resource management; California Delta

## 1. Introduction

Evapotranspiration (ET) describes the rate of exchange of water vapor between the land surface and the atmosphere, including water directly evaporated from the soil, open water, and other surfaces (E), as well as water consumed and transpired by plants in the process of biophysical development (T). The ability to accurately estimate ET spatially and temporally over large areas is critical to a broad range of applications, for example, in managing water resources, in assessing agricultural water use, and in monitoring ecosystem health and the impacts of drought [1]. Demands for real-time access to daily ET information at spatial scales from field to globe are ever increasing in support of food and water security applications [2]. To address these data needs will require timely and accurate methods for mapping ET based on modeling and/or remote sensing, and accurate estimates of forcing variables to serve as the model input.

The primary factor governing ET in many cases is the solar radiation load, which largely determines the energy available at the land surface to evaporate water [3,4]. Solar energy drives the surface energy balance: the partitioning of net (incoming minus outgoing) longwave and shortwave radiation primarily between sensible heat, latent heat (equivalent to ET, but in energy units), and soil heat conduction flux. Without an accurate estimate of the incoming shortwave radiation, the ability of any ET model to predict evaporative fluxes is severely limited. In addition to insolation, ET is modulated by soil moisture availability; vegetation amount, structure, and health; and meteorological conditions such as air temperature, vapor pressure deficit, and wind speed. An ET model with broad utility will consider each of these factors.

Given the critical importance of the solar radiation input, it is informative to assess the impact of insolation data sources on the accuracy and utility of operational ET methods. In this study, we investigate this in the context of a multi-scale surface energy balance algorithm based on remote sensing measurements of land-surface temperature (LST). The Atmosphere-Land Exchange Inverse (ALEXI) model and associated flux disaggregation algorithm (DisALEXI) use LST data derived from satellite-based thermal infrared (TIR) imaging systems to map ET and other surface energy fluxes at resolutions of 30 m at a landscape scale to 5 km at continental to global scales. ALEXI/DisALEXI datasets are being used for applications in drought monitoring, irrigation management, yield prediction, and in investigating changes in water-use accompanying landcover and land use change [5–9]. The choice of insolation product used in these applications will have ramifications for both model accuracy and operational feasibility.

In this study, we investigated the impact of different sources of solar global irradiance (insolation) data in an application of ALEXI/DisALEXI over central California, focusing on the Sacramento–San Joaquin River Delta region. This area was the focus of an ET model intercomparison study assessing the utility of remotely sensed consumptive use estimates in informing the response to California's Sustainable Groundwater Management Act (SGMA) [10]. ALEXI/DisALEXI is also being used to develop irrigation management strategies for viticulture as part of the Grape Remote sensing and Atmospheric Profile and Evapotranspiration eXperiment (GRAPEX [11]). Both applications have specific requirements regarding data latency, spatial and temporal resolution, period of record, accuracy, and reliability of data delivery. Continental to global scale applications will have additional requirements regarding the data domain coverage.

Here, we build on the baseline ET results presented by Anderson et al. [12] driven by modeled insolation from the Climate Forecast System Reanalysis (CFSR) global dataset generated at hourly timesteps and at a 0.25° spatial resolution. We compare the CFSR insolation to hourly satellite-based insolation datasets developed over the continental United States (CONUS) at 0.2°, using data from the Geostationary Operational Environmental Satellites (GOES) and the coarser resolution (1°) global Clouds and the Earth's Radiant Energy System (CERES) satellite-based insolation product. The impact of the insolation data source on ET retrievals at daily to yearly timescales is also assessed, with the purpose of identifying the optimal inputs for different water resource applications.

## 2. Materials and Methods

*2.1. Study Domain*

The study domain is shown in Figure 1 and is consistent with that used in the study presented in Anderson et al. [12]. This region includes the California Delta region at the confluence of the Sacramento and San Joaquin Rivers, which serves as a major hub for water supply within the state of California. Medellín-Azuara et al. [10] describe an ET model intercomparison study conducted over the CA Delta, commissioned by the Delta Water Master. Timeseries of daily ET data were produced with an ensemble of ET models for the water years 2015–2016 and intercompared to assess the level of agreement between the models, as well as the variability in model behavior for different landcover types. ALEXI/DisALEXI timeseries generated using CFSR insolation inputs for the ET model intercomparison study were further assessed by Anderson et al. [12] in comparison with observations collected at multiple flux towers within the modeling domain, sampling various representative landcovers, including vineyards, alfalfa, rice, and wetlands. The general locations of these towers are indicated in Figure 1. These include towers deployed on Sherman and Twitchell Islands to monitor water and carbon fluxes from a chronosequence of restored wetlands and converted rice fields in comparison with dryland agriculture [13]. Additionally, towers associated with the GRAPEX [11] study site in the Borden Ranch viticultural area outside of Lodi, CA were used in the assessment.

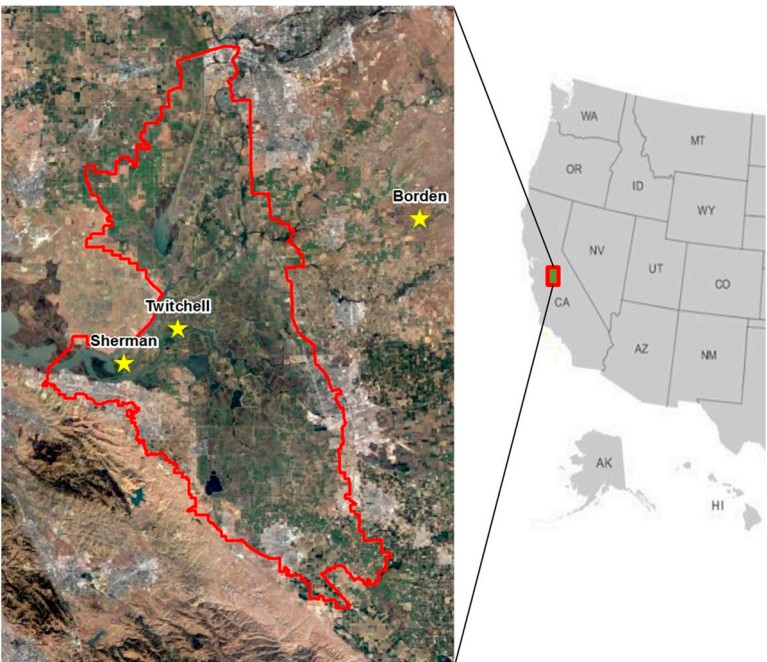

**Figure 1.** Study domain covering the Sacramento–San Joaquin Delta in California's Central Valley. The Legal Delta Area is delineated in red, with flux tower sites on Twitchell and Sherman Islands and in vineyards in the Borden Ranch area indicated with yellow stars.

The study period covered the water years of 2015 and 2016 (WY15-16), with a water year running from October 1 to September 30. This period came near the end of the extended severe drought (2012–2017) in California, which precipitated conversions in cropping and irrigation practices within the Central Valley. In this paper, the results of Anderson et al. [12] were revisited to assess the dependence of ET retrieval accuracy on the accuracy of the insolation data used to drive the energy balance model.

*2.2. ET Remote Sensing Framework*

2.2.1. ALEXI/DisALEXI

Surface energy fluxes were computed over the study domain using the Atmosphere-Land Exchange Inverse (ALEXI) surface energy balance model and the associated flux disaggregation technique, DisALEXI. This modeling system is described in Anderson et al. [12] and prior studies (e.g., [14–16]) and is only briefly summarized here. Land-surface exchanges in ALEXI/DisALEXI are governed by the Two-Source Energy Balance (TSEB) parameterization described originally by Norman et al. [17] with improvements in Kustas et al. [18,19]. In the TSEB, the soil and canopy energy budgets (designated with subscripts 's' and 'c', respectively) are solved separately:

$$
\begin{aligned}
RN_C &= H_C + \lambda E_C \\
RN_S &= H_S + \lambda E_S + G \\
RN &= H + \lambda E + G
\end{aligned}
\tag{1}
$$

where $RN$ is net radiation, $H$ is sensible heat, $\lambda E$ is latent heat, and $G$ is the flux of heat into the substrate below the canopy. Sensible heat flux from the vegetation canopy ($H_C$) and substrate ($H_S$-typically soil) combine in series to form the net flux $H$, as constrained by the component temperature estimates $T_C$ and $T_S$ and the above-canopy air temperature $T_A$. The component temperatures are extracted via the system of model equations from radiances inferred by the bulk directional surface radiometric temperature:

$$
T_{RAD}(\theta)^4 = f(\theta)T_C{}^4 + [1 - f(\theta)]T_S{}^4
\tag{2}
$$

where $f(\theta)$ and $T_{RAD}(\theta)$ are the directional vegetation cover fraction and radiometric temperature at the view angle of the thermal sensor, $\theta$. Net radiation is computed using an analytical canopy light interception formulation described in Campbell and Norman [20] and Anderson et al. [21]. The treatment of solar irradiance follows the methods described in Weiss and Norman [22] for partitioning observations of incoming shortwave radiation into visible and near infrared direct beam and diffuse components.

For spatially distributed implementation of the TSEB over a landscape, a time-differential approach can be used to replace boundary conditions in near-surface air temperature $T_A$, which are difficult to prescribe with adequate accuracy and minimal bias with respect to the remotely sensed $T_{RAD}$ data. The regional ALEXI modeling framework applies the TSEB at two times during the morning hours (approximately an hour after sunrise and an hour before local noon), with energy closure over this interval provided by a simple atmospheric boundary layer (ABL) growth model [23]. As early as 1993, Diak and Whipple [24] demonstrated the strong linkage between ABL growth and land-surface heating. In that study, an ABL model was used to diagnose values of sensible heating, but only at rawinsonde locations. ALEXI works well for the same reason, based on its ability to diagnose the time-change of the boundary layer height and its relation to sensible heat. However, the parameterization and the use of forecast model initial conditions allows ALEXI to be run at any location where time-changes of land-surface temperature, solar energy values, and initial atmospheric conditions are available, not just at rawinsonde locations as in Diak and Whipple [24].

Model surface temperature inputs at the two times are typically acquired via geostationary satellites, and sensitivity tests show that the model is sensitive to the morning change in $T_{RAD}$, but is minimally sensitive to time-invariant absolute errors in $T_{RAD}$ retrieval [23]. In this approach, $T_A$ is diagnosed at the boundary between the surface and ABL models rather than being prescribed as a model input. Daily (24-h) integrated latent heat flux is computed by scaling the $\lambda E$ derived at the second $T_{RAD}$ observation time (pre-noon) using the local solar radiation curve (see Section 2.2.2). The daily ET ($ETd$; mm·d$^{-1}$) can be obtained from the daily latent heat flux ($\lambda Ed$; MJ·m$^{-2}$·d$^{-1}$) using the latent heat of vaporization ($\lambda$, approximately 2.45 MJ·m$^{-2}$ to evaporate 1 mm of water at 20 °C).

The spatial resolution of ALEXI flux estimates is constrained by the resolution of the high temporal frequency LST observation source (here, geostationary satellites)—generally on the order of several km. For generating higher resolution ET maps, capable of resolving individual crop fields or land management units, an ALEXI flux disaggregation approach (DisALEXI) was developed as described in References [25,26]. In the DisALEXI step, the TSEB is executed over a gridded model domain using higher resolution LST retrievals from Landsat (30 m, spatially sharpened as described in Section 2.3.1) or MODIS (1 km native or 500 m sharpened) with an initial $T_A$ boundary derived from meteorological analyses. This nominal boundary condition is then iteratively adjusted at the ALEXI pixel scale until the DisALEXI *ETd* field reaggregates to the ALEXI baseline flux [14]. At the final step in the iteration, the modified $T_A$ field is spatially smoothed to remove boxy ALEXI-scale artifacts in the output flux maps.

### 2.2.2. Upscaling to Daily ET Using Insolation

In the TIR-based version of ALEXI and DisALEXI, direct retrievals of ET are obtainable only on days when the skies are clear at the acquisition times of the satellite images used (for ALEXI, the full time period between morning acquisitions is required to be clear). In the current approach, the instantaneous ET at the overpass time was upscaled to a 24-h flux by conserving the ratio of ET to insolation flux, following recommendations by Cammalleri et al. [27]. That study compared several potential flux metrics for upscaling, including ratios with available energy, net radiation, and reference ET, and found that the best performance over a range of flux sites was obtained using insolation. Ryu et al. [28] also found insolation to be a robust scaling factor using data from the global flux tower datasets sampling multiple biomes. Here, the ratio ($f_{SUN}$) of instantaneous ET (*ETi*) to insolation (*Rsi*) flux was computed at the satellite overpass time:

$$f_{SUN} = ETi/Rsi \tag{3}$$

and then the daily ET flux (*ETd*) was estimated from the 24-h integrated insolation (*Rsd*) for that day:

$$ETd = f_{SUN} \times Rsd \tag{4}$$

In ALEXI, which is typically run using surface temperature data acquired by geostationary satellites with high temporal frequency (15-min), a Savitsky–Golay filter is first applied to the $f_{SUN}$ timeseries at the pixel level to minimize day-to-day variability due to undetected cloud contamination, and then the smoothed timeseries is gap-filled using linear or spline interpolation. ALEXI *ETd* is then reconstructed using Equation (4) applied to the daily insolation maps. An all-sky retrieval system is under development using LST information derived from microwave Ka-band data (several km resolution), which can see through clouds, circumventing the need for gap filling at the ALEXI scale [29]. To obtain optimal correspondence with the ALEXI timeseries, MODIS retrievals (near daily overpass), a similar temporal smoothing and gap-filling procedure is applied to the ratio of MODIS to ALEXI *ETd*.

### 2.2.3. Data Fusion

Landsat overpasses are generally too infrequent (8–16 days or longer, depending on cloud cover and the number of concurrently operating Landsat platforms) to justify the kind of temporal gap-filling described in Section 2.2.2. Instead, a data fusion methodology has been employed to fuse Landsat (high spatial/low temporal resolution) and MODIS (low spatial/high temporal resolution) ET timeseries into a single *ETd* "datacube", with daily timesteps and 30-m pixels. The Spatial and Temporal Adaptive Reflectance Fusion Model (STARFM [30]) compares MODIS and Landsat image pairs on days when both are available, computes spatial weighting statistics, and then applies these weights to downscale MODIS images between clear-sky Landsat overpasses. STARFM was originally developed to fuse surface reflectance imagery, but it has also been applied successfully to ET datasets developed over a variety of agricultural and forested landscapes in the U.S. [6,7,15,16,31,32] and in Spain [33].

In this data fusion strategy, the direct Landsat ET retrievals on clear Landsat overpass dates serve as tie points in the temporal reconstruction. Time behavior between these clear-date retrievals is governed by changes in the MODIS ET at a coarser scale. Therefore, the accuracy of the reconstructed daily datacube is largely sensitive to the magnitude of the solar radiation estimates on these clear Landsat days, and secondarily to variations in the daily insolation between overpasses.

*2.3. Insolation Datasets*

Model inputs to the ET data fusion system are described in detail by Anderson et al. [12]. Here, we focus on examining three insolation datasets that could be used as input to ALEXI/DisALEXI and other ET mapping approaches, representing major classes of reanalysis and remotely sensed solar radiation products available today.

2.3.1. CFSR

Meteorological inputs of air temperature (used as the initial $T_A$ boundary condition in the DisALEXI iteration), vapor pressure, atmospheric pressure, and wind speed were obtained from the near-real-time CFSR at a 0.25° resolution [34,35] and at 3-h intervals with global coverage, maintained by the National Centers for Environmental Prediction (NCEP) (College Park, MD, USA). CFSR also generates hourly gridded insolation data at a 0.25° resolution, which were used in the ET timeseries retrievals for the CA Delta described by Anderson et al. [12]. In CFSR, shortwave radiation transfer through the atmosphere is modeled using methods described by Chou et al. [36] and Hou et al. [37]. Near-real-time CFSR data are available from the NCEP with a latency of around six hours (http://nomads.ncep.noaa.gov/pub/data/nccf/com/cfs/prod/cfs). Retrospective CFSR data are available from NOAA's National Centers for Environmental Information (NCEI) from 1979 to present (https://www.ncdc.noaa.gov/data-access/model-data/model-datasets/climate-forecast-system-version2-cfsv2).

2.3.2. SSEC

To contrast with the coarse-scale CFSR modeled insolation, we also tested an hourly satellite-based insolation product generated at a 0.2° resolution by the University of Wisconsin Space Science and Engineering Center (SSEC) (Madison, WI, USA) using visible data from the GOES-East and GOES-West platforms [38]. The insolation model runs in the computational environment of the Man-computer Interactive Data Access System (McIDAS [39]), developed and distributed by the SSEC. Near-real-time insolation fields from this system have been used for regional- and continental-scale land surface carbon and water flux assessments [5,26,40–42], solar energy spatial and temporal variability analyses [43], subsurface hydrologic modeling efforts, and currently for agricultural forecasting products from the University of Wisconsin-Madison Extension (Madison, WI, USA) [44].

The radiative transfer (RT) models used to estimate insolation from the GOES data are simple, physically-based, and depend upon calibration (digital counts to energy) of the GOES-visible sensors (Figure 2). The basic equation set has been presented in Diak et al. [45], but was modified from the absorption coefficient form to the calculation of transmittances for the various atmospheric processes similar to Reference [46]. With knowledge of the solar constant, date, and time, the model calculates how much solar energy is entering the earth-atmosphere system for any latitude–longitude location. The role of the satellite measurement is to estimate how much of this incident energy escapes back to space. Subsequently, the atmospheric RT model, based on principles of energy conservation, is used to partition the energy remaining in the earth-atmosphere system into its various components, importantly surface insolation. Separate models for the clear and cloudy atmosphere are employed. A detailed description of the insolation model is provided in Diak [38].

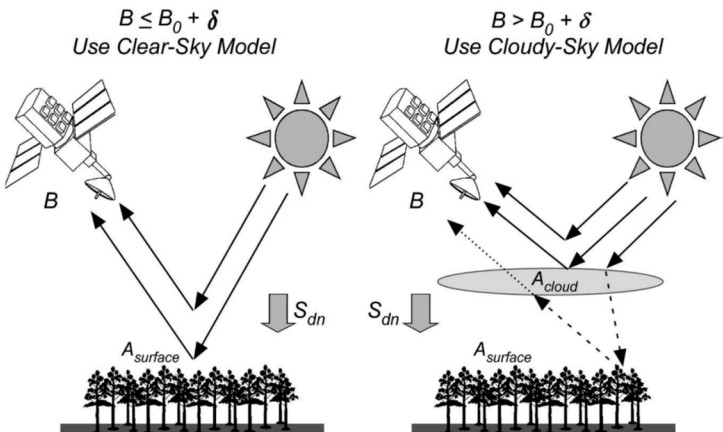

**Figure 2.** Graphical depiction of the physical model employed for (left-hand side) clear-sky conditions and (right-hand side) cloudy-sky conditions; B refers to the brightness observed by the satellite, while $B_0$ is the clear-air brightness threshold; Sdn refers to the downward shortwave radiation flux; and $A_{surface}$ and $A_{cloud}$ refer to the surface and cloud albedos, respectively.

Despite its relative simplicity, recent studies have found the quality of results from this simple model to be similar to that from more complex formulations. Paech et al. [47] used the simple model as a basis for evaluating evapotranspiration in Florida and found the model insolation statistics to be similar to results from the NOAA satellite-based insolation model. The insolation results presented in this study were also similar in quality to the results for detailed for the NOAA model by Wonsick et al. [48]. In Diak [38], seasonal results of this insolation model were evaluated against in-situ pyranometer measurements at 45 sites of the United States Climate Research Network (USCRN) (Washington, DC, USA). Accuracy for hourly insolation estimates was 15% to 20%, while the accuracy range for the daily integrated insolation total ranged from 5% to 10%. Calibration of the visible channel has been an issue for the GOES satellites; their sensitivity degrades significantly with time and up until GOES-16 (brought on-line on 1 January 2018) there was no on-board calibration—it was done vicariously. The insolation model for the period described here used calibration coefficients provided by NOAA, and a Real Earth™ display of the prior day's insolation can be viewed at https://www.ssec.wisc.edu/insolation/.

Half-hourly and daily-integrated insolation estimates for the previous day in ASCII text format are available on the SSEC anonymous ftp site (ftp://prodserv1.ssec.wisc.edu) in the subdirectory insolation_high_res for no charge. The half-hourly data from the GOES-East and -West were downloaded and consolidated onto a single 0.2° grid covering the CONUS. These datasets have periodic gaps in coverage due to satellite outages or issues with model development, which were filled using CFSR to generate fully filled data grids.

### 2.3.3. CERES

For comparison with the CONUS GOES-based SSEC product, we also tested a global 1° insolation product developed by NASA's Langley Research Center using data from the CERES satellite instruments. The CERES Synoptic Radiative Fluxes and Clouds (SYN) product at a spatial resolution of 1° uses 3-h geostationary and MODIS radiances, cloud properties, MODIS aerosol observations, and atmospheric profiles provided by the NASA Global Modeling and Assimilation Office (GMAO) model to more accurately model the diurnal variability between CERES observations [49,50]. One disadvantage with the CERES SYN product is that it is not available in near-real-time and it usually has a latency in availability on the order of 3 to 6 months. Retrospective products are available at https://ceres.larc.nasa.gov/products.php?product=SYN1deg.

*2.4. Flux Datasets*

Micrometeorological data from eight eddy covariance (EC) systems operating within the study domain were used by Anderson et al. [12] to evaluate the fused CFSR-based WY15-16 ET timeseries over different landcover types (Table 1). Two flux towers were located in the Borden Ranch viticultural area north of Lodi, CA in adjacent vineyards with Pinot noir vines of age ~12 years (Lodi1) and 9 years (Lodi2) during the experiment timeframe. These are operated as part of the GRAPEX experiment. Also operating during the study period were four AmeriFlux EC towers installed on Twitchell Island, at the core of the Delta region. These towers include two sites in water-intensive crops: rice (US-Twt; doi:10.17190/AMF/1246140) and alfalfa (US-Tw3; doi:10.17190/AMF/1246149). Two additional towers are sited in restored wetlands: the East End wetland (US-Tw4; doi:10.17190/AMF/1246151), restored in 2014, is largely vegetated with some open water channels, while the West Pond site (US-Tw1; doi:10.17190/AMF/1246147) dates back to 1997, and the vegetation is now fully closed. Two AmeriFlux tower sites on nearby Sherman Island were also used, sampling the mature Mayberry wetland (US-Myb; doi:10.17190/AMF/1246139) established in 2010, and the sparsely vegetated US-Sne wetland (doi:10.17190/AMF/1418684), converted from pasture and newly flooded in December of 2016. Details regarding the Twitchell and Sherman Island measurements are provided by Eichelmann [13].

**Table 1.** Flux towers used in the analysis.

| Site | Tower | Name | Cover | Latitude | Longitude |
|---|---|---|---|---|---|
| Borden Ranch | Lodi1 | | vineyard | 38.2894 | −121.1178 |
| Borden Ranch | Lodi2 | | vineyard | 38.2805 | −121.1176 |
| Twitchell Island | US-Tw1 | West Pond | old wetland | 38.1073 | −121.6468 |
| Twitchell Island | US-Tw4 | East End | young wetland | 38.1028 | −121.6413 |
| Twitchell Island | US-Twt | | rice | 38.1087 | −121.6530 |
| Twitchell Island | US-Tw3 | | alfalfa | 38.1151 | −121.6468 |
| Sherman Island | US-Myb | Mayberry | intermediate wetland | 38.0498 | −121.7650 |
| Sherman Island | US-Sne | Sherman | new wetland | 38.0369 | −121.7547 |

All the flux sites, except US-Myb, were instrumented with pyranometers to measure the incoming shortwave radiation (solar radiation) at 30-min intervals. For the Lodi towers, insolation measurements were obtained with the incoming shortwave component of a Kipp & Zonen CNR-1 four-component net radiometer. The two towers were located less than 1 km apart and daily totals agreed to within 2%, with an $R^2$ of 0.99. Twitchell and Sherman Island shortwave radiation was measured using the pyranometer component of Hukseflux NR01 four-way net radiometers, except at the Twitchell rice site, where a Kipp & Zonen CM11 pyranometer was employed. In a detailed comparison experiment conducted in Fall 2016, the radiation sensors for all Twitchell and Sherman Island sites were deployed at the same location next to the Twitchell rice measurement tower for three weeks. All instruments measuring incoming shortwave radiation, regardless of the time deployed in the field or make and model, showed very close agreement with measurements within 5% of each other. When compared to a recently factory calibrated 'golden standard', all sensors had $R^2$ values above 0.986 with regression slopes between 0.972 and 1.002.

The EC technique is known to yield turbulent flux estimates that do not fully close the observed energy budget, yielding closure errors given by $1 - (H + \lambda E)/(RN - G)$ typically on the order of 10–20% or higher in some cases [51,52]. For comparison with model estimates, which assume closure, observed latent heat fluxes over non-wetland surfaces have been closed using the residual of the energy balance. For the wetland sites, closure has not been attempted due to uncertainties in measuring the heat storage, *G*, within the water substrate given the dynamic bathymetry and fluctuating water tables. For those sites, observed $\lambda E$ was increased by a nominal 10% to account for flow distortion effects associated with non-orthogonal sonic anemometer configurations in the EC systems used here as in References [53–56].

## 3. Results

### 3.1. Insolation Product Evaluation at the Flux Tower Sites

Representative maps of daily insolation over CONUS for day of year (DOY) 200 (19 July) in 2017 from the CFSR, CERES, and SSEC insolation datasets are shown in Figure 3. While general cloud features are consistent between the datasets, the maps differ significantly in terms of spatial resolution and fidelity of structure. CFSR clearly reflect a more artificial, model-based treatment of cloud processes, while the satellite inputs to SSEC and CERES generate more realistic structures albeit at very different spatial resolutions.

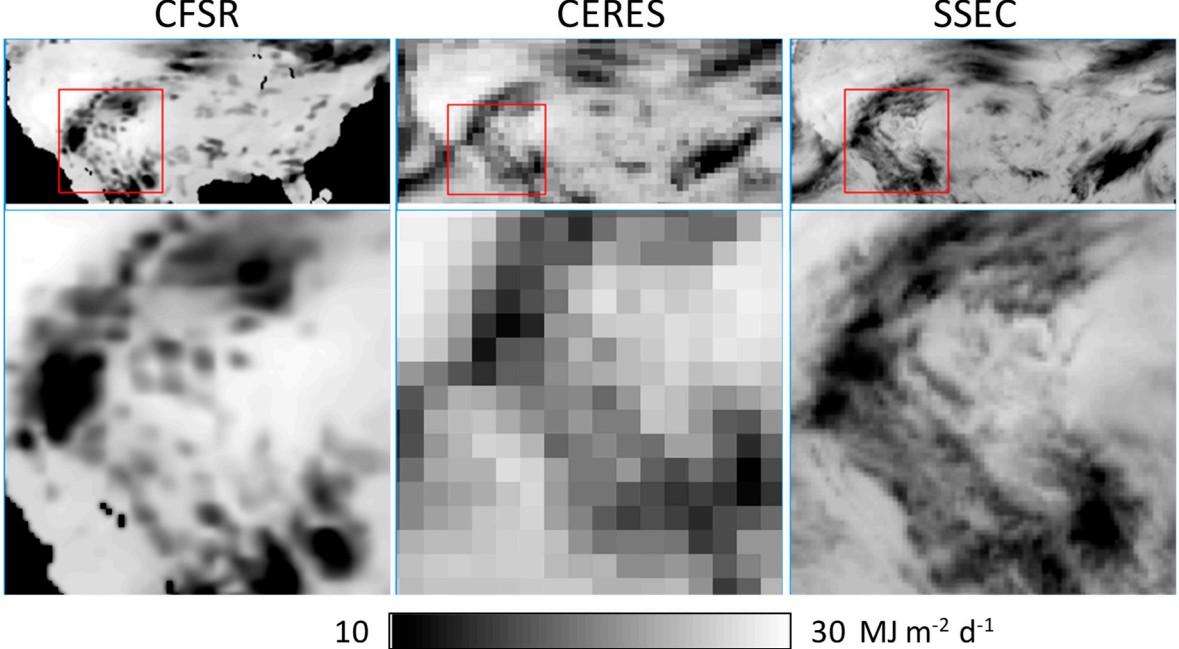

**Figure 3.** Comparison of daily insolation maps for 2017 DOY 200 from the CFSR, CERES, and SSEC products. A zoom in over the southwestern U.S. is included to highlight differences in structure and resolution between the products.

Table 2 provides statistical metrics comparing insolation estimates from the three products with fluxes observed at the flux sites equipped with pyranometers, as well as for all the sites combined. These metrics included the mean observed flux (<O>), mean bias error (MBE) in model minus observations, root mean square error (RMSE), Nash–Sutcliff coefficient of efficiency (NSE [57]), coefficient of determination ($R^2$), mean absolute error (MAE), and the relative error (RE = MAE/<O>). Graphical representations of MAE, RE, and MBE at daily to annual timesteps are shown in Figure 4, while Figure 5 shows scatter plots versus observations at daily timesteps for each site collecting pyranometer data.

At daily timesteps, the SSEC and CERES satellite-based products had significantly lower errors in comparison with the CFSR, with the RE reducing from approximately 0.09 to 0.05. However, the MBE was lower with CFSR at daily to annual timesteps. This may have resulted from better calibration of the clear-sky upper envelope in the CFSR over this part of the CONUS in comparison with the satellite products, which must additionally account for annual degradation in the imaging systems. Of the two satellite products, SSEC generally had a lower RE but marginally higher bias over this timeframe in comparison with CERES. (Note that some of the tower sites did not have a full annual record of insolation measurements. These missing site-years resulted in a sign flip in the MBE in CFSR insolation at the yearly timestep.)

**Table 2.** Statistical metrics of insolation dataset performance at flux sites on clear Landsat overpass dates (LS day) and at daily, weekly, monthly, and yearly timesteps, reported in units of MJ·m$^{-2}$·d$^{-1}$.

| | | | | CFSR | | | | | | CERES | | | | | | SSEC | | | | | |
|---|---|---|---|---|---|---|---|---|---|---|---|---|---|---|---|---|---|---|---|---|---|
| Timescale | Tower | N | <O> | MBE | RMSE | NSE | R2 | MAE | RE | MBE | RMSE | NSE | R2 | MAE | RE | MBE | RMSE | NSE | R2 | MAD | RE |
| DAILY | Lodi1 | 761 | 19.05 | 0.34 | 2.92 | 0.87 | 0.89 | 1.84 | 0.097 | 0.37 | 1.86 | 0.95 | 0.95 | 1.13 | 0.059 | 0.16 | 1.33 | 0.97 | 0.97 | 0.76 | 0.040 |
| | Lodi2 | 740 | 19.54 | 0.09 | 2.99 | 0.86 | 0.88 | 1.86 | 0.095 | −0.10 | 1.35 | 0.97 | 0.97 | 0.79 | 0.040 | −0.10 | 1.35 | 0.97 | 0.97 | 0.79 | 0.040 |
| | US-Sne | 162 | 24.72 | −0.08 | 1.78 | 0.94 | 0.95 | 1.07 | 0.043 | −1.18 | 1.60 | 0.95 | 0.99 | 1.37 | 0.056 | −1.18 | 1.60 | 0.95 | 0.99 | 1.37 | 0.056 |
| | US-Tw1 | 687 | 19.49 | 0.40 | 2.57 | 0.91 | 0.92 | 1.68 | 0.086 | 0.03 | 1.41 | 0.97 | 0.98 | 1.13 | 0.058 | 0.03 | 1.41 | 0.97 | 0.98 | 1.13 | 0.058 |
| | US-Tw3 | 797 | 18.99 | −0.21 | 2.68 | 0.91 | 0.92 | 1.60 | 0.084 | −0.46 | 1.48 | 0.97 | 0.98 | 1.07 | 0.056 | −0.46 | 1.48 | 0.97 | 0.98 | 1.07 | 0.056 |
| | US-Tw4 | 762 | 20.24 | −0.53 | 2.74 | 0.90 | 0.91 | 1.63 | 0.080 | −0.82 | 1.63 | 0.96 | 0.98 | 1.23 | 0.061 | −0.82 | 1.63 | 0.96 | 0.98 | 1.23 | 0.061 |
| | US-Twt | 792 | 19.26 | 0.12 | 2.66 | 0.90 | 0.91 | 1.65 | 0.086 | −0.18 | 1.29 | 0.98 | 0.98 | 0.83 | 0.043 | −0.18 | 1.29 | 0.98 | 0.98 | 0.83 | 0.043 |
| LS DAY | ALL | 365 | 23.39 | 0.38 | 1.98 | 0.90 | 0.91 | 1.17 | 0.050 | 0.04 | 1.40 | 0.95 | 0.95 | 0.90 | 0.038 | −0.31 | 1.15 | 0.97 | 0.97 | 0.90 | 0.038 |
| DAILY | ALL | 4701 | 19.61 | 0.02 | 2.74 | 0.90 | 0.91 | 1.69 | 0.086 | −0.23 | 1.52 | 0.97 | 0.97 | 1.04 | 0.053 | −0.27 | 1.43 | 0.97 | 0.98 | 0.98 | 0.050 |
| WEEKLY | ALL | 661 | 19.76 | 0.03 | 1.41 | 0.97 | 0.97 | 1.04 | 0.052 | −0.26 | 1.08 | 0.98 | 0.99 | 0.82 | 0.042 | −0.29 | 1.03 | 0.98 | 0.99 | 0.78 | 0.039 |
| MONTHLY | ALL | 151 | 20.03 | 0.08 | 0.94 | 0.99 | 0.99 | 0.77 | 0.038 | −0.27 | 0.97 | 0.98 | 0.99 | 0.75 | 0.038 | −0.31 | 0.95 | 0.98 | 0.99 | 0.72 | 0.036 |
| YEARLY | ALL | 11 | 19.26 | −0.62 | 0.89 | −0.58 | 0.20 | 0.73 | 0.038 | −0.77 | 1.00 | −1.01 | 0.16 | 0.83 | 0.043 | −0.81 | 1.05 | −1.18 | 0.13 | 0.87 | 0.045 |

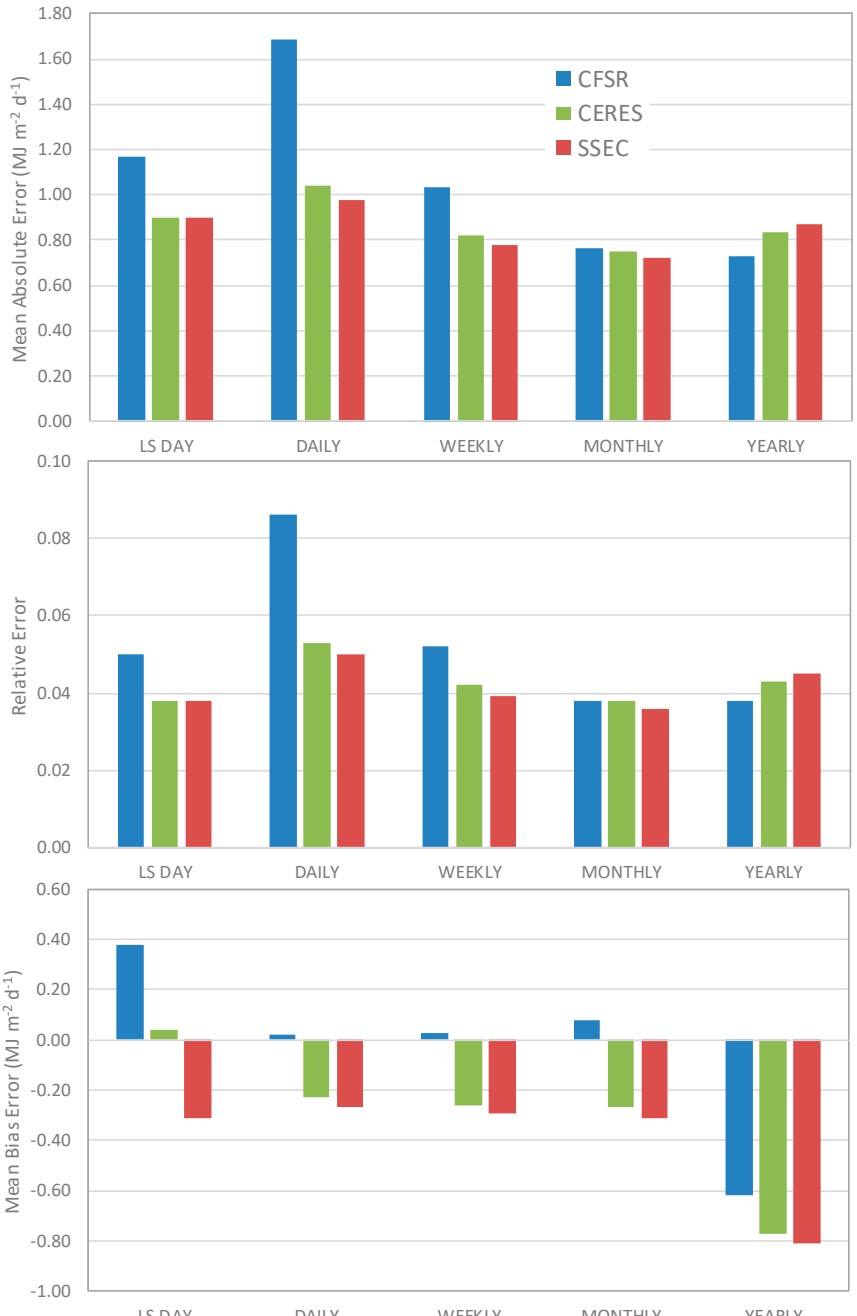

**Figure 4.** Mean absolute error (MAE), relative error (RE), and mean bias error (MBE) in insolation products on clear Landsat overpass dates (LS DAY) and at daily, weekly, monthly, and yearly timesteps over the study period.

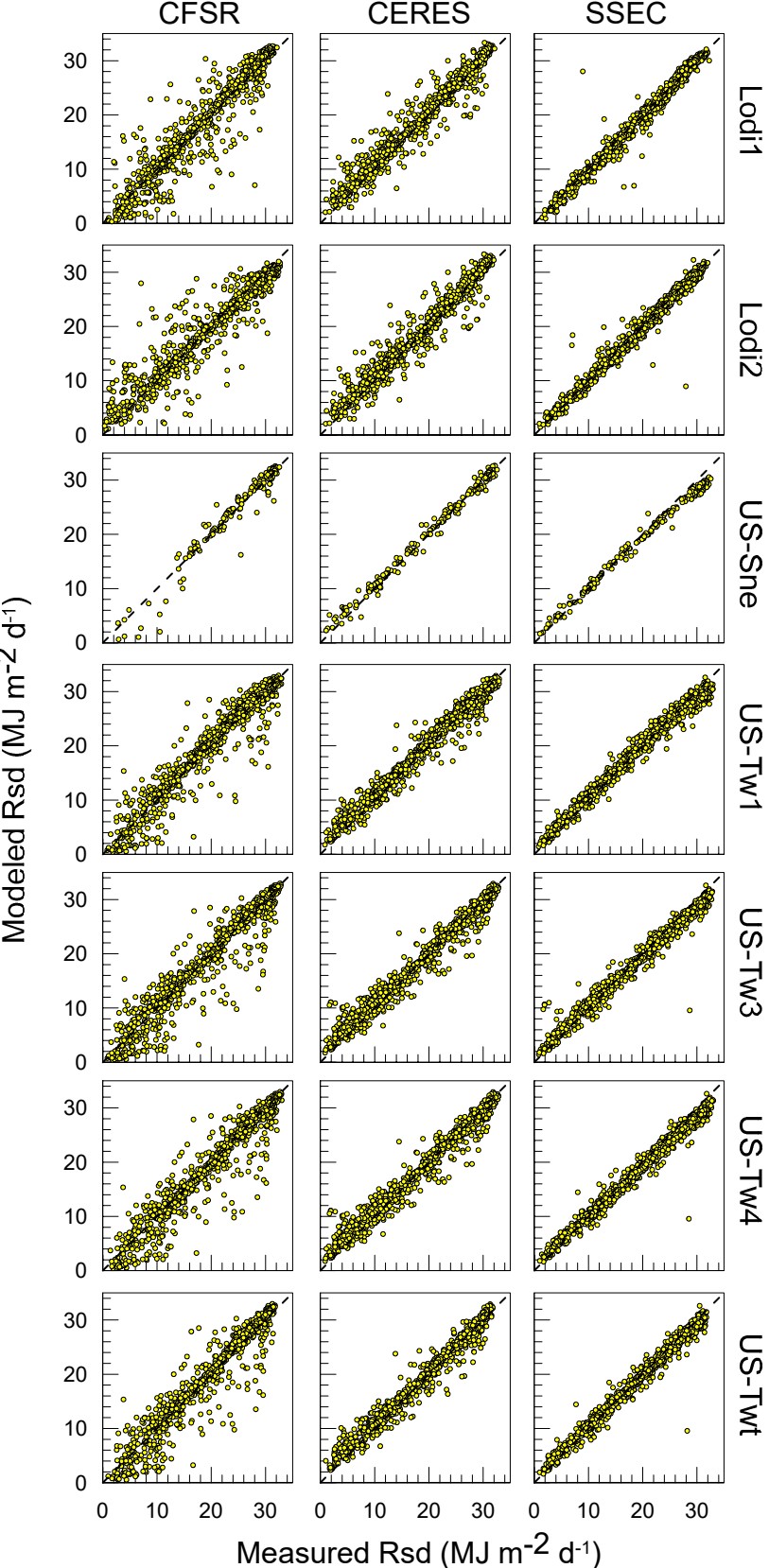

**Figure 5.** Comparison of the daily insolation fluxes from the CFSR, SSEC, and CERES datasets with pyranometer observations at the flux sites within the study region.

Timeseries comparisons at two sites showing significant improvement in the satellite versus modeled insolation (Lodi2 and US-Twt) further illuminated the error characteristics of the three insolation datasets (Figure 6). All three captured the predominantly cloudless conditions past mid-2016, a virtually rain-free period which served to exacerbate drought conditions in the Central Valley. CFSR tended to overpredict cloud effects (also see scatter plots in Figure 5), while SSEC best captured the strength and timing of cloud impacts on incident solar radiation at the scale of observation.

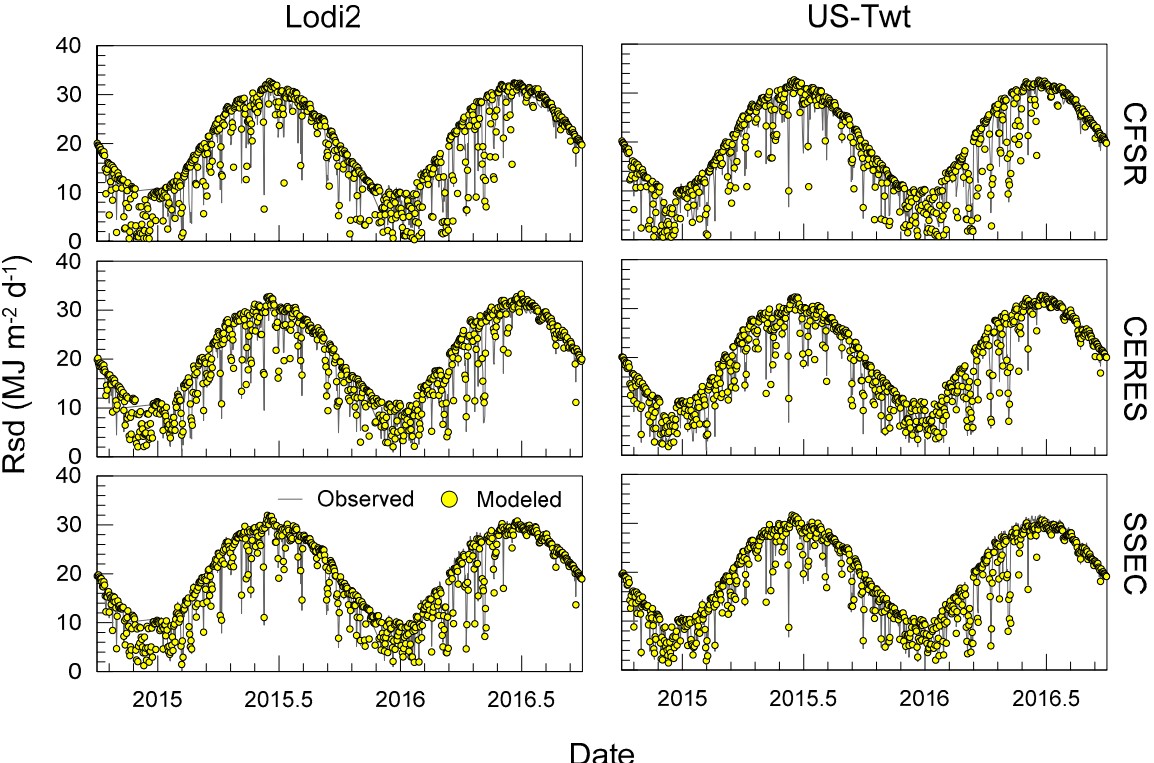

**Figure 6.** Daily modeled and observed insolation timeseries at the Lodi2 and US-Twt flux sites.

*3.2. Energy Balance on Landsat Dates*

In comparison with the daily results, insolation product performance, in terms of the MAE and RE, was more similar between the CFSR and the satellite insolation datasets on the Landsat overpass days when conditions were clear at the time of overpass and a direct retrieval of ET could be obtained (Figure 4 and Table 2). CFSR tended to overestimate daily insolation on these days, while SSEC underestimated by a similar amount on average over the two water years. CERES had minimal bias on clear Landsat overpass days. As noted in Section 2.2.3, these days are critical to the full daily reconstruction as these direct Landsat ET retrievals define key tie points in the data fusion process.

Scatter plots of modeled versus measured insolation from the CFSR and SSEC datasets on clear Landsat overpass dates are shown in Figure 7, along with the derived net radiation and partitioning between sensible, latent, and soil heating diagnosed with DisALEXI. These two insolation products were selected as a demonstration to bracket the range of expected DisALEXI performance. Agreement with observations for all flux components was marginally improved using the SSEC insolation product, primarily due to the reduction of outliers. While MAE in insolation was reduced from 1.2 to 0.9 $MJ \cdot m^{-2} \cdot d^{-1}$ moving from CFSR to SSEC, respectively, the impact on net radiation was less apparent, with a reduction from 1.6 to 1.5 $MJ \cdot m^{-2} \cdot d^{-1}$. Latent heat (and ET, in units of mass) performance on Landsat dates was similar for each of the insolation datasets, with a RE of 0.19 in both cases and a MAE of 1.8 $MJ \cdot m^{-2} \cdot d^{-1}$ (0.75 $mm \cdot d^{-1}$). The lack of improvement in the ET retrievals on Landsat dates using SSEC insolation was consistent with the reasonable capability of the model reanalysis to capture insolation fluxes on predominantly clear days.

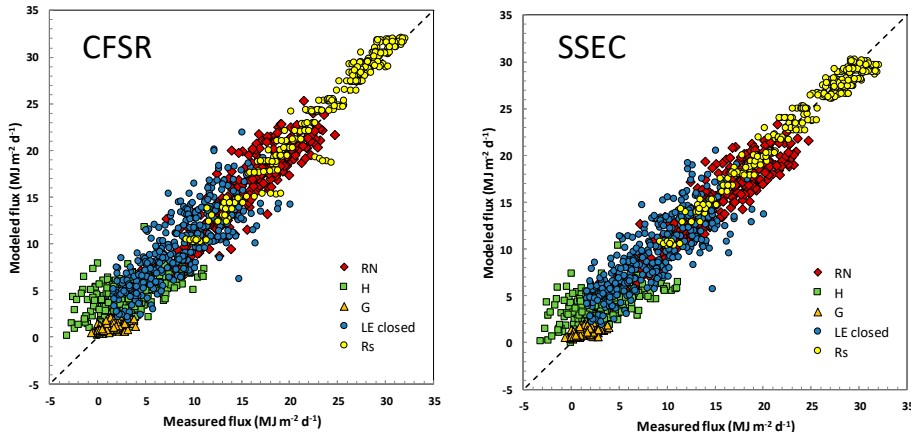

**Figure 7.** Energy flux partitioning generated with DisALEXI on clear Landsat dates using CFSR and SSEC insolation data as input compared to the tower flux observations.

### 3.3. Daily ET from Data Fusion

Statistics describing the performance of the full daily ET timeseries generated via the Landsat-MODIS data fusion process are provided in Table 3 at daily timesteps and aggregated to weekly, monthly, and annual averages, and visually summarized in Figure 8. These metrics elucidate the impact of the insolation data source on the accuracy of daily ET retrievals from the data fusion system. While there was some improvement in ET at the daily timestep using the SSEC insolation, with the MAE decreasing from 0.72 with CFSR to 0.69 mm d$^{-1}$ (RE of 0.23 to 0.22), the impact was not large—particularly given the large improvement in accuracy in the daily insolation drivers (RE of 0.09 to 0.05). As demonstrated in Figure 9, showing scatter plot comparisons of the measured and modeled ET for each tower site, the effect of the satellite insolation was primarily to reduce the magnitude of outliers, and the degree of improvement varied from site to site. The value of the satellite-based dataset diminished for applications at weekly and longer timescales, where the performance of the two insolation datasets was more similar (Figure 8).

The largest improvements in ET reconstruction using the SSEC insolation occurred at the Lodi2 (vineyard) site where the MAE decreased from 0.52 to 0.43 mm d$^{-1}$, and at US-Twt (rice) with a MAE reduction from 0.91 to 0.83 mm d$^{-1}$ (Table 3). Assessment of the ET timeseries at US-Twt demonstrated the nature of the improvement in performance—primarily a more realistic portrayal of day-to-day ET variability with the SSEC-based retrievals, particularly on cloudy days (Figure 10). Given the quality of the SSEC product, residual ET errors must be due to errors in other inputs or in the modeling assumptions.

**Table 3.** Statistical metrics of ET retrieval performance (mm d$^{-1}$) at flux sites on clear Landsat overpass dates (LS Day) and at daily, weekly, monthly, and yearly timesteps using insolation inputs from the CFSR and SSEC datasets.

| Timescale | Tower | N | <O> | CFSR | | | | | | SSEC | | | | | |
| | | | | MBE | RMSE | NSE | R2 | MAE | RE | MBE | RMSE | NSE | R2 | MAE | RE |
|---|---|---|---|---|---|---|---|---|---|---|---|---|---|---|---|
| DAILY | Lodi1 | 450 | 3.44 | 0.11 | 0.80 | 0.76 | 0.77 | 0.63 | 0.184 | 0.10 | 0.76 | 0.78 | 0.79 | 0.60 | 0.175 |
| | Lodi2 | 438 | 3.35 | 0.10 | 0.67 | 0.75 | 0.77 | 0.52 | 0.156 | 0.07 | 0.60 | 0.80 | 0.80 | 0.45 | 0.135 |
| | US-Myb | 818 | 3.11 | 0.10 | 0.81 | 0.84 | 0.85 | 0.63 | 0.203 | 0.07 | 0.75 | 0.86 | 0.86 | 0.58 | 0.188 |
| | US-Sne | 161 | 3.70 | −0.31 | 0.84 | 0.53 | 0.74 | 0.70 | 0.190 | −0.40 | 0.82 | 0.56 | 0.71 | 0.65 | 0.176 |
| | US-Tw1 | 818 | 3.03 | −0.11 | 0.99 | 0.79 | 0.79 | 0.80 | 0.265 | −0.13 | 0.98 | 0.79 | 0.80 | 0.78 | 0.258 |
| | US-Tw3 | 817 | 2.77 | −0.39 | 1.09 | 0.56 | 0.64 | 0.81 | 0.291 | −0.42 | 1.08 | 0.57 | 0.65 | 0.78 | 0.281 |
| | US-Tw4 | 818 | 3.57 | 0.16 | 0.77 | 0.90 | 0.90 | 0.60 | 0.169 | 0.12 | 0.77 | 0.90 | 0.90 | 0.60 | 0.167 |
| | US-Twt | 817 | 2.96 | 0.48 | 1.24 | 0.74 | 0.80 | 0.91 | 0.306 | 0.44 | 1.14 | 0.78 | 0.82 | 0.83 | 0.281 |
| LS DAY | ALL | 330 | 3.90 | 0.04 | 0.96 | 0.74 | 0.75 | 0.75 | 0.192 | −0.07 | 0.94 | 0.75 | 0.75 | 0.75 | 0.191 |
| DAILY | ALL | 5241 | 3.10 | 0.08 | 0.95 | 0.78 | 0.79 | 0.72 | 0.233 | 0.05 | 0.91 | 0.80 | 0.80 | 0.69 | 0.221 |
| WEEKLY | ALL | 720 | 3.18 | 0.09 | 0.76 | 0.84 | 0.85 | 0.57 | 0.179 | 0.05 | 0.75 | 0.85 | 0.85 | 0.56 | 0.174 |
| MONTHLY | ALL | 169 | 3.19 | 0.08 | 0.58 | 0.90 | 0.91 | 0.44 | 0.137 | 0.04 | 0.57 | 0.90 | 0.91 | 0.42 | 0.132 |
| YEARLY | ALL | 10 | 3.02 | 0.07 | 0.28 | 0.02 | 0.64 | 0.24 | 0.078 | 0.04 | 0.28 | 0.02 | 0.63 | 0.23 | 0.076 |

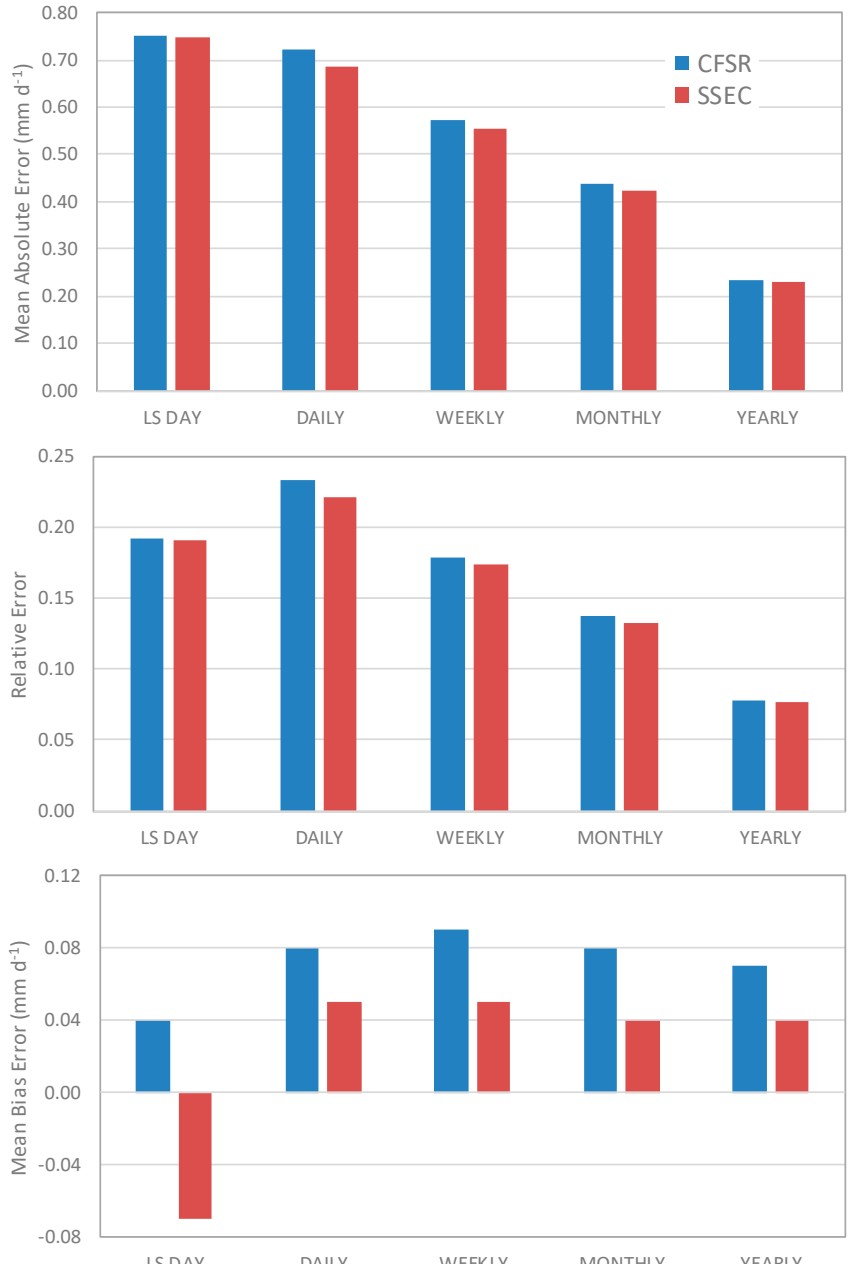

**Figure 8.** MAE, RE, and MBE in ET retrievals using the CFSR and SSEC insolation inputs on clear Landsat overpass dates (LS DAY) and at daily, weekly, monthly, and yearly timesteps over the study period.

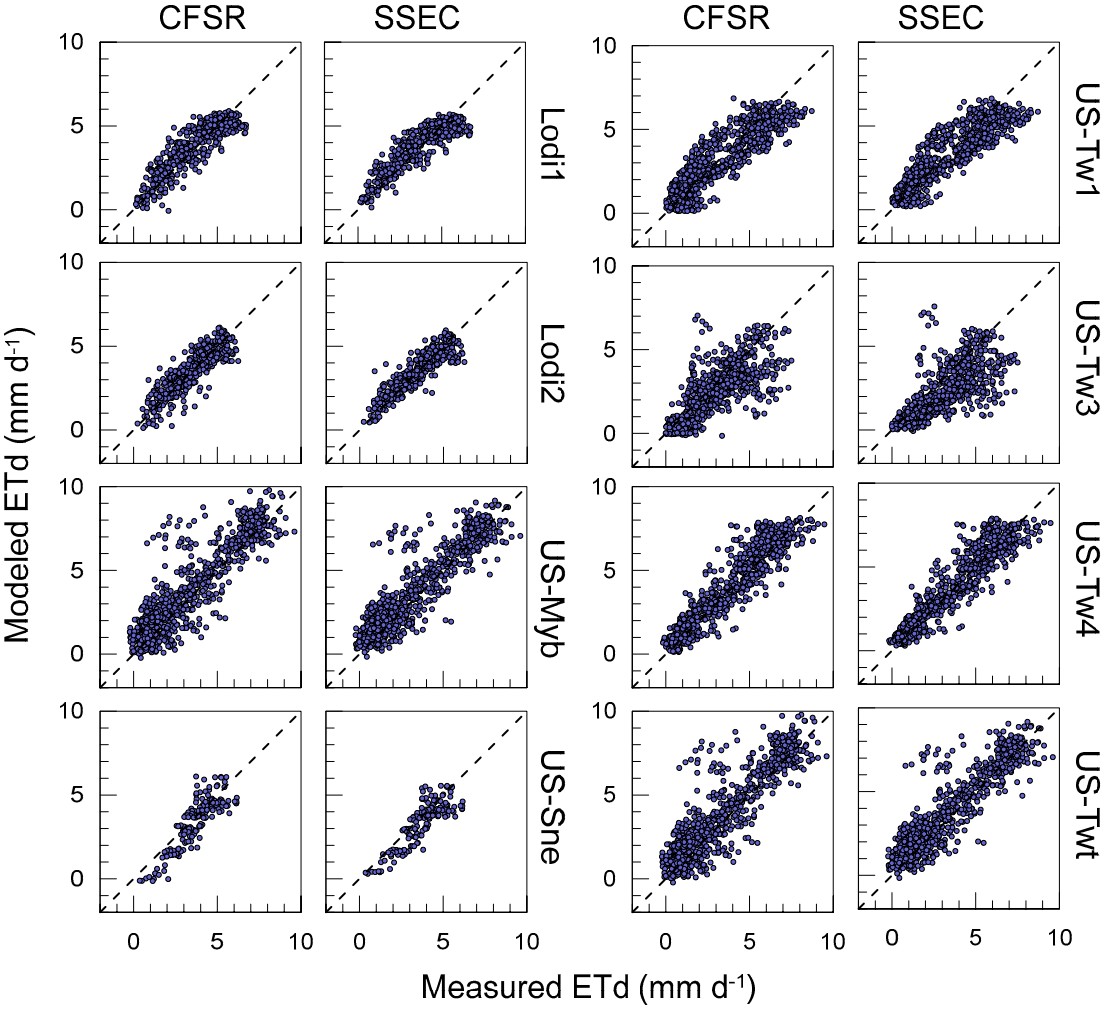

**Figure 9.** Comparison of daily observed and modeled ET fluxes generated using the CFSR and SSEC insolation datasets.

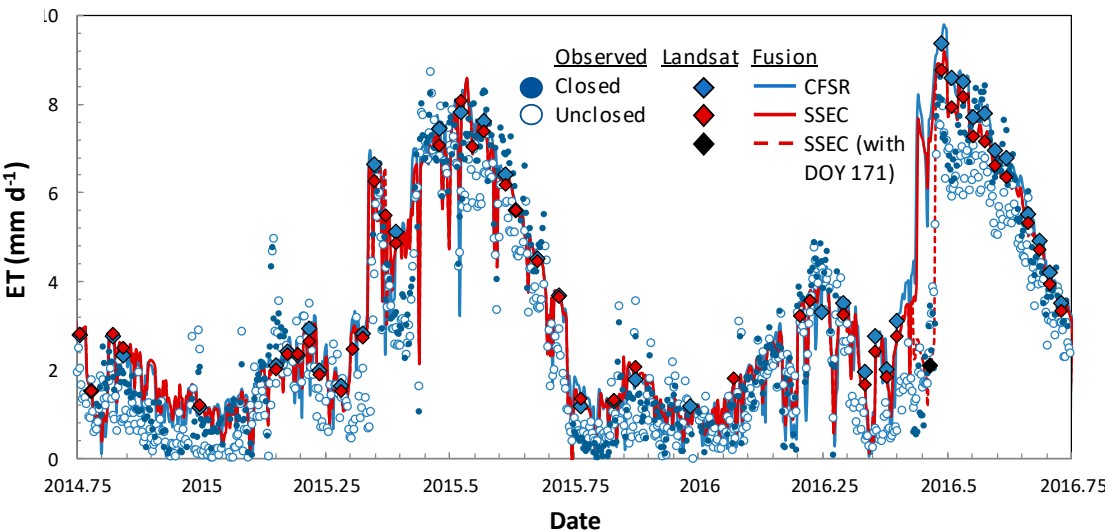

**Figure 10.** Timeseries of daily ET observations (unclosed and closed circles) at US-Twt and retrievals using the CFSR (light blue) and SSEC (red) insolation inputs (diamonds: on Landsat dates, solid lines: fused timeseries). The fused timeseries generated using the Landsat retrieval on DOY 171 (black diamond) is also shown (red dotted line).

In general, the tower sites with the largest ET retrieval errors (US-Twt, US-Tw3, and US-Tw1) tended to be in landcovers where ET was more decoupled from the solar radiation load as evidenced in the flux tower observations (Figure 11). In the densely vegetated West Pond wetland (US-Tw1), increases in spring ET were delayed relative to the solar radiation curve due to a dense mat of litter material that inhibited new vegetative growth [13]. In the case of US-Twt (rice) and US-Tw3 (alfalfa), this decoupling was due to management activities, such as regulated flooding/drainage (rice) and periodic cuttings (alfalfa) [12,13]. Some portion of this high temporal resolution structure in water use phenology was missed by the ET data fusion system, leading to higher errors at these sites that were not resolved with the improved insolation inputs. For the rice site, a critical Landsat scene on DOY 171 in 2016 was omitted from the fusion process due to clouds and contrails in the southern part of the domain that were not captured by the Landsat cloud mask. Including this scene, which occurred at an inflection in moisture dynamics just prior to reflooding of the rice paddy, reduced the RMSE in ETd at US-Tw3 from 1.14 to 0.97 mm d$^{-1}$ at the daily timescale using the SSEC insolation inputs (Figure 11), and from 0.91 to 0.88 mm d$^{-1}$ for all the sites combined. In this case, the excess error was in part due to inadequate temporal sampling by Landsat during times of rapid change occurring at spatial scales too small to be well-captured in the MODIS timeseries. Landsat temporal sampling also poses a challenge in reproducing observed fluxes over alfalfa, where monthly cuttings may not coincide with a Landsat overpass and subsequent vegetation regrowth is rapid.

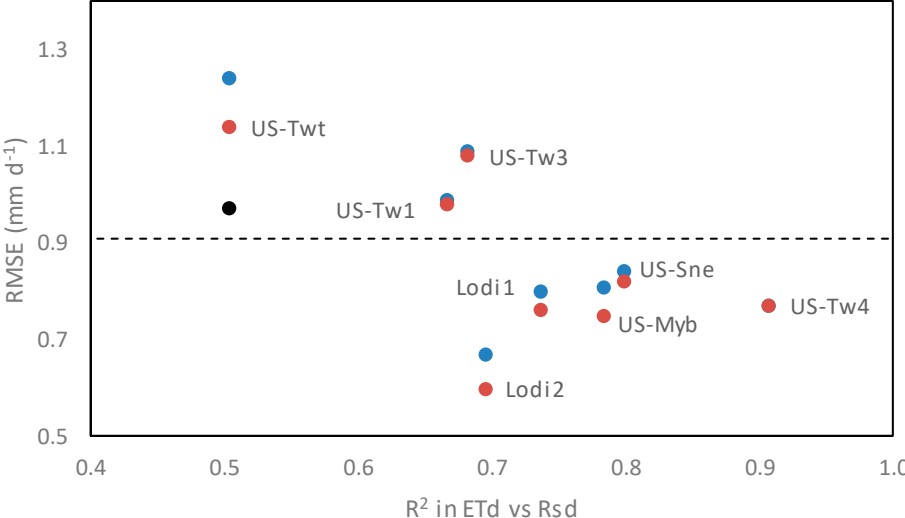

**Figure 11.** Root mean square error (RMSE) in daily ET retrievals using the CFSR (blue dots) and SSEC (red dots) inputs vs. the R$^2$ value computed between daily solar radiation and ET (unclosed) observed at the tower sites. The dotted line indicates the RMSE obtained with all sites combined. Also shown is the RMSE at US-Twt from a fusion experiment including the Landsat ET retrieval on DOY 171 in 2016 (black dot), omitted from the standard run due to extensive cloud cover in the southern part of the modeling domain.

Heterogeneous conditions with the tower footprint scale impacted model-measurement agreement at the US-Tw3 and US-Tw1 sites. The US-Tw3 site was highly variable in terms of stand health, and eddy covariance footprint analyses show off-field contributions may influence the tower fluxes [13]. The restored wetland sampled by the US-Tw1 tower is quite narrow (about 90 m wide near the tower—a few Landsat pixels). In both cases, off-site contributions to the tower fluxes were likely and a rigorous daily footprint analysis would be required to optimize the selection of model pixels for comparison with tower fluxes (see also Figure 4 in [12]).

## 4. Discussion

### 4.1. How Representative Are These Results?

The results presented here may be specific to the study site in Central California, which is dominated in the summer months by clear conditions under which the CFSR performance is optimal and clear-sky Landsat retrievals are frequent, facilitating quick correction in the fused ET timeseries if an individual retrieval is poor. A more significant improvement in ET accuracy will likely be obtained in regions with higher climatological frequency of cloud cover, such as in the northwest and eastern U.S. In these regions, fewer direct ET retrievals will be available, serving as tie points in the fusion reconstruction. Enhanced errors in the CFSR representation of insolation under clouds will further degrade performance between clear Landsat overpasses. A global intercomparison of multiple reanalysis insolation products, including CFSR, with ground-based observations and with the CERES satellite product revealed significant spatiotemporal patterns in bias and the RMSE due in part to biases in the cloud fraction and model treatment of aerosols [58].

Work is underway to intercompare CFSR and SSEC-based ET timeseries at other flux sites distributed across the U.S. to draw generalized conclusions regarding the impact of insolation dataset on ET retrieval accuracy.

### 4.2. Which Is the Optimal Insolation Datasource?

Selection of an optimal insolation data source for ET mapping depends on many factors and may vary by application and region of interest. The three datasets investigated here have distinct advantages and disadvantages.

CFSR has obvious advantages for operational use, including low data latency (available next day), hourly timesteps, complete and global coverage, and a long period of records generated with a consistent processing system. The latter is particularly important for climatological and trend analyses, which can be very sensitive to discontinuous changes in processing of inputs [59]. Disadvantages include lower accuracy at daily timesteps, artificial spatial structures, significant bias in some regions, and much lower spatial resolution in comparison with the ET model pixel scale.

In contrast, the spatial resolution for the SSEC GOES product is comparable to the ALEXI pixel scale. The precision obtained over the CA Delta sites was excellent, and the product is freely available with low data latency (available next day) and at half-hourly timesteps. Importantly, in ALEXI the solar radiation and thermal inputs are optimally consistent, both spatially and temporally, when both are derived from imagery from the same geostationary platform. Disadvantages are limitations in spatial domain (currently CONUS only), frequent gaps that require filling, and moderate bias and shifts in calibration, e.g., due to new satellites and sensor degradation. There has been no reprocessing for a consistent long-term archive.

CERES, despite its relatively coarse spatial resolution, showed comparable RMSE to the SSEC product over the flux sites in central CA, and lower bias errors. At hourly timesteps and with complete global coverage, this dataset holds significant promise for retrospective ET mapping over a range of spatial scales. Unfortunately, at present, the latency in data delivery (6-month lag) makes CERES insolation unsuitable for real-time applications.

## 5. Conclusions

This study examined the impacts of insolation data source on a multi-sensor TIR remote sensing data fusion approach for mapping actual evapotranspiration at field scale and daily timesteps. Pyranometer observations collected in central California for water years 2015–2016 were compared to data from three gridded insolation datasets, including the Climate Forecast System Reanalysis (CFSR), and geostationary satellite-based products over the CONUS (SSEC) and the globe (CERES). Daily relative errors in the two satellite datasets (RE = 0.05) were approximately half that from the

reanalysis data (RE = 0.09), resulting from the improved capacity to capture cloud occurrence and impact on surface insolation.

The resulting improvement in ET retrievals at these tower sites was less notable using the satellite-based insolation inputs, with a RE of 0.23 and 0.22 for CFSR and SSEC at daily timesteps, respectively, and a RMSE of 0.95 and 0.91 mm d$^{-1}$. The lack of significant improvement in ET performance suggests that other modeling inputs or errors dominate over the set of sites examined here. This may be due in part to the relatively clear-sky conditions prevalent in central California during the peak growing season. CFSR errors are smaller under the clear-sky conditions when direct satellite retrievals can be conducted. Residual model errors in runs using the high quality SSEC insolation inputs are largest at tower sites where the observed ET is less coupled to solar radiation forcings, such as in highly managed agricultural fields (rice and alfalfa) and in a wetland system where storage fluxes and dense vegetation residue delay spring ET increases. These sites also tended to be spatially complex, with significant off-site contributions to tower flux measurements necessitating rigorous daily footprint analyses to better clarify the spatial model performance. A greater improvement in ET retrievals using satellite-based insolation may be expected in regions where evaporative fluxes are energy limited over a greater portion of the growing season.

The optimal insolation data source depends on the data latency requirements and spatial extent of the application in question. Both CFSR and SSEC datasets are available in near-real-time (less than one day latency); however, SSEC coverage currently includes only the continental U.S. While the global CERES insolation product has error statistics similar to that of SSEC, the latency is currently close to 6 months, precluding real-time use. Over the California Delta and climatologically similar regions in the western U.S., the CFSR data may suffice for real-time ET modeling efforts.

**Author Contributions:** Conceptualization: M.A., G.D., F.G.; Data Fusion system: F.G., C.H., Y.Y., M.A., Image Processing: G.D., K.K., M.A., Data collection: D.B., E.E., K.H., W.K.; Formal analysis: M.A., G.D., D.B., E.E., K.H., K.K. Writing—original draft: M.A., G.D.; Writing—review and editing: D.B., E.E., K.H., W.K., K.K.

**Funding:** This work was funded in part by the NASA MEASURES and Land Cover Land Use Change programs and the NASA ECOSTRESS Earth Ventures Instrument project [80NSSC18K0483].

**Acknowledgments:** The authors would like to acknowledge the contribution of UC Berkeley Biomet lab technician Daphne Szutu in conducting an extensive radiometer comparison experiment at the Twitchell Rice site.

**Conflicts of Interest:** The authors declare no conflict of interest.

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
