# Peer review of "Impact of Insolation Data Source on Remote Sensing Retrievals of Evapotranspiration over the California Delta"

_remotesensing, doi:10.3390/rs11030216_

Round 1
Reviewer 1 Report
Remote Sensing.
Manuscript number: 401281 26 November 2018
Title: “Impact of Insolation Data Source on Remote Sensing Retrievals of Evapotranspiration over the California Delta ”
Authors: Martha Anderson1,*, George Diak2, Feng Gao1, Kyle Knipper1, Christopher Hain3, Elke 5 Eichelmann4, Kyle S. Hemes4, Dennis Baldocchi4, William Kustas1, and Yun Yang1 6
1 USDA-ARS, Hydrology and Remote Sensing Laboratory, Beltsville, MD 20705, USA;
Abstract:
The manuscript investigates the practical utility of three different insolation datasets within the context of a satellite-based remote sensing framework for mapping ET at high spatiotemporal resolution, in an application over Sacramento-San Joaquin Delta region in California. The datasets tested include one reanalysis product: the Climate System Forecast Reanalysis (CFSR) at 0.25O spatial resolution, and two remote sensing insolation products generated with geostationary satellite imagery: a product for the continental United States at 0.2O developed by the University of Wisconsin Space Sciences and Engineering Center (SSEC) and a coarser resolution (1O) global Clouds and the Earth’s Radiant Energy System (CERES) product.
Recommendations:
1-Abstract is very long. Information less relevant should be deleted, Thank you.
2- Keywords should include the place where the study takes place.
3- Table 1. Longitude and latitude data have four decimal figures. this number of Figures is high. The number of figures should be reduced in two.
4-What variable is “insolation”???. It is known that the three fundamental components of solar radiation at the Earth´s surface are: solar global irradiance, diffuse solar irradiance and direct solar irradiance. Sunshine hours also gives information about solar resource assessment. Please clarify. Thank you.
5-Figure 5 contains 21 panels (21 fit lines). Each one should inform about equation, number of data used, correlation coefficient. The same for Figure 9 results, please.
6- Figure 6. The figure compares measured and modelled solar irradiation values. But this comparison should be done by scatter plot, by the statistical estimators or errors (mabe, mbe, remse, etc). and by cumulative frequency curves. In Figure 6 is difficult to distinguish observed and modelled values.
Conclusions: The manuscript should be corrected following the recommendations. Chiefly, the manuscript should clarify, the name and meaning of the variables that are using along the manuscript.
Author Response
We thank the reviewer for their recommendations, and have implemented them as described below in red text.
Recommendations:
1-Abstract is very long. Information less relevant should be deleted, Thank you.
The abstract has been shortened by about 100 words.
2- Keywords should include the place where the study takes place.
“California Delta” has been added to the keywords.
3- Table 1. Longitude and latitude data have four decimal figures. this number of Figures is high. The number of figures should be reduced in two.
We prefer to retain the extra digits to provide a location accuracy comparable to the 30m Landsat grid scale.
4-What variable is “insolation”???. It is known that the three fundamental components of solar radiation at the Earth´s surface are: solar global irradiance, diffuse solar irradiance and direct solar irradiance. Sunshine hours also gives information about solar resource assessment. Please clarify. Thank you.
By insolation we mean solar global irradiance. This is now clarified in the text.
5-Figure 5 contains 21 panels (21 fit lines). Each one should inform about equation, number of data used, correlation coefficient. The same for Figure 9 results, please.
We feel that this additional text will be visually distracting and will necessarily obscure some of the points plotted. This information is provided in Tables 2 and 3 – the figures are included give a visual sense of the level of agreement.
6- Figure 6. The figure compares measured and modelled solar irradiation values. But this comparison should be done by scatter plot, by the statistical estimators or errors (mabe, mbe, remse, etc). and by cumulative frequency curves. In Figure 6 is difficult to distinguish observed and modelled values.
Scatter plots of these comparisons are provided in Fig. 5 with statistical metrics in Table 2. Figure 6 is included to give the reader a sense of the temporal nature of the errors in each insolation data source. While the clear sky upper bound in the observed fluxes is not apparent, the fact that it is obscured by the modeled points suggests the models do well in that limit. The reader can also see where the model does or does not agree with observations on cloudy days.
An error was found in Fig. 6 – the right-hand panels had plotted data from a different station. This error has been corrected in the revised manuscript.
Conclusions: The manuscript should be corrected following the recommendations. Chiefly, the manuscript should clarify, the name and meaning of the variables that are using along the manuscript.
Reviewer 2 Report
Comments can be found in the annotated pdf file uploaded in Remote Sensing journal platform.

Author Response
We thank the reviewer for the careful editorial reading and comments embedded in the pdf attachment. These suggestions have been implemented in the revised text.